# Trends in Antimicrobial Resistance of Uropathogens Isolated from Urinary Tract Infections in a Tertiary Care Hospital in Dhaka, Bangladesh

**DOI:** 10.3390/antibiotics13100925

**Published:** 2024-09-27

**Authors:** Sara Sadia Chowdhury, Promi Tahsin, Yun Xu, Abu Syed Md. Mosaddek, Howbeer Muhamadali, Royston Goodacre

**Affiliations:** 1Centre for Metabolomics Research, Department of Biochemistry, Cell and Systems Biology, Institute of Systems, Molecular and Integrative Biology, University of Liverpool, Biosciences Building, Crown Street, Liverpool L69 7ZB, UK; yun.xu@liverpool.ac.uk (Y.X.); howbeer.muhamad-ali@liverpool.ac.uk (H.M.); roy.goodacre@liverpool.ac.uk (R.G.); 2Department of Pharmacology, Uttara Adhunik Medical College (UAMC), Sonargaon Janapath, Uttara, Dhaka 1230, Bangladeshdrmosaddek1968@gmail.com (A.S.M.M.)

**Keywords:** antimicrobial resistance, urinary tract infection, uropathogens, *E. coli*, *Klebsiella*, *Enterococcus*, multidrug resistance, prevalence, Bangladesh

## Abstract

**Background/Objectives**: Urinary tract infection (UTI) is a prevalent microbial infection in medical practise, leading to significant patient morbidity and increased treatment costs, particularly in developing countries. This retrospective study, conducted at a tertiary care hospital in Dhaka, Bangladesh, aims to examine the antimicrobial resistance (AMR) patterns of uropathogens and evaluate whether these patterns are influenced by demographic factors such as gender, age, or patient status. **Methods**: Standard microbiological techniques were used to identify uropathogens, and AMR patterns were determined using the Kirby–Bauer disc diffusion method. **Results**: Out of 6549 urine samples, 1001 cultures were positive. The infection was more prevalent in females compared to males. The incidence of UTIs in children aged 0–10 years accounted for 12.59% of the total cases, with this age group also exhibiting the highest rate of polymicrobial infections. Among the bacterial uropathogens, 71.19% of isolates were multidrug resistant (MDR) and 84.27% were resistant to at least one antibiotic. *Escherichia coli* (*n* = 544, 73.90% MDR) and *Klebsiella* species (*n* = 143, 48.95% MDR) were the most common Gram-negative uropathogens, while *Enterococcus* species (*n* = 78, 94.87% MDR) was the predominant Gram-positive isolate in this study. Our results indicate that most uropathogens showed resistance against ceftazidime, followed by cefuroxime, trimethoprim-sulfamethoxazole, amoxicillin-clavulanate, and netilmicin. Moderate levels of resistance were observed against ciprofloxacin, levofloxacin, aztreonam, and cefpodoxime. **Conclusions**: Amikacin was observed to be effective against Gram-negative uropathogens, whereas cefixime was more active against Gram-positive microorganisms, such as *Enterococcus* species. Moreover, a principal coordinate analysis (PCoA) depicted no significant influence of gender, patient status, or age on AMR patterns. For the continued usefulness of most antibiotics, periodic analysis of the AMR patterns of uropathogens can help assess the rise of MDR bacteria, and therefore guide the selection of appropriate antibiotic treatment strategies.

## 1. Introduction

Urinary tract infections (UTIs) are caused by the invasion of microorganisms in the urinary tract, including the urinary bladder, prostate, kidney, or collecting system, and the condition may range from asymptomatic bacteriuria to perinephric abscess with sepsis [1,2,3]. They are considered to be the second-most prominent type of bacterial infections, affecting 150 million people annually around the globe, and account for up to 35% of all nosocomial infections [1,4,5]. UTI patients are effectively managed by identifying the causative pathogen of the infection first, and then selecting the effective antimicrobial agent against it [6].

Recently, the rising rates of resistance among bacterial uropathogens have become a major public health crisis worldwide. *Escherichia coli* is the principal uropathogen, accounting for 75 to 90% of bacterial isolates in uncomplicated UTIs [6,7,8,9]. Additionally, *Klebsiella* spp. are predominant pathogens in both UTIs and nosocomial infections, and easily tend to become multidrug resistant [6]. Several major factors associated with the increase in antimicrobial resistance (AMR) includes the misuse of antimicrobials [10], using broad-spectrum antimicrobials frequently that alter the intestinal flora, and the inappropriate dosage and duration of treatment [11,12]. This complicates the treatment of UTIs by increasing the rate of treatment failure and patient morbidity, costs of reassessment and retreatment, and the use of broad-spectrum antibiotics [9]. In a developing country like Bangladesh, a substantial number of patients with UTI obtain antibiotics straight from community pharmacies without a prescription or consulting with a medical professional [13]. Moreover, regional clinicians commonly diagnose microbial infections based on clinical judgement and prescribe broad-spectrum antibiotics on an empirical basis prior to the laboratory results of urine culture being available. These factors unfavourably affect the sensitivity pattern of microbes, potentially leading to the development and spread of AMR strains [14,15,16].

Yet, most AMR surveillance studies do not consider the situation of every geographical location in the world. Only extensive country-specific (and indeed, in some cases, regional) surveillance studies will provide information about the causative pathogens of UTIs and their AMR patterns, which will further aid clinicians in selecting the correct antimicrobial empirical treatment [13]. Therefore, this study was conducted to determine the AMR patterns of bacterial isolates from patient urine samples in 11 commonly prescribed antibiotics.

## 2. Results

### 2.1. Prevalence of UTI According to Age and Gender

In this study, a total of 6549 patients suspected of having a UTI were analyzed, comprising 2392 males and 4157 females, resulting in a male-to-female ratio of 1:1.74. Among them, 326 males and 675 females tested positive for a UTI. The overall infection rate was observed to be higher in females (16.25%) compared to males (13.62%), as shown in Table 1. Figure 1 presents the UTI rates for both genders across different age groups and polymicrobial UTI rates within these age brackets. A higher incidence of infection was noted among females aged 20 to 70 years, with the infection rate peaking in the age group of 50 to 60 years. In contrast, for males, the infection rate was high in the 50 to 80 years age range, with men aged 60 to 70 years showing the greatest prevalence. Regardless of gender, the elevated incidence of UTIs among children aged 0 to 10 years was noticeable. Furthermore, our study shows the polymicrobial infection rate was the highest in children, followed by people aged 60 to 70 years. In general, individuals aged 50 to 70 years predominantly tested positive for UTIs, with a marked decline in infection rates observed in those over 70 years of age.

### 2.2. Microbiological Profile of Uropathogens

Microbiological cultures of 6549 urine samples detected 1001 cases of UTI (Figure 2). Among these, 833 (83.22%) were bacterial infections, 58 (5.79%) were fungal infections (*Candida* spp.), and 110 (10.99%) were polymicrobial UTIs (Appendix A). Among the Gram-negative bacterial isolates, the following were identified: *E. coli* (544 isolates; 54.35%), *Klebsiella* spp. (143 isolates; 14.29%), *Enterobacter* spp. (34 isolates; 3.40%), *Pseudomonas* spp. (nine isolates; 0.90%), *Proteus* spp. (eight isolates; 0.80%), *Acinetobacter* spp. (five isolates; 0.50%), *Acetobacter* spp. (two isolates; 0.20%), *Citrobacter* spp. (one isolate; 0.10%), and *Morganella morganii* (one isolate; 0.10%). In contrast, the Gram-positive bacterial isolates included *Enterococcus* spp. (78 isolates; 7.79%), *Staphylococcus aureus* (seven isolates; 0.70%), and *Lactobacillus* spp. (one isolate; 0.10%). Overall, the majority of the bacterial uropathogens were Gram-negative (89.68%), with *E. coli* being predominant, followed by *Klebsiella* spp. and *Enterobacter* spp. Among Gram-positive bacteria (10.32%), *Enterococcus* spp. was the most common uropathogen.

### 2.3. Antibiotic Resistance Patterns and Trends in Bacterial Uropathogens

Out of 833 unimicrobial bacterial uropathogens, 593 (71.19%) were identified as multidrug resistant (MDR) (Table 2). Among these, *E. coli* demonstrated an MDR prevalence of 73.90%, *Klebsiella* spp. 48.95%, *Enterobacter* spp. 70.59%, and *Enterococcus* spp. 94.85%. Figure 3 and Appendix A present the resistance profiles of these uropathogens to 11 antibiotics, while their resistance trends over time (months) are detailed in Appendix A.

#### 2.3.1. *E. coli*

*E. coli* shows over 50% resistance to several antibiotics, including cefuroxime (64.71%), ceftazidime (64.71%), amoxicillin–clavulanate (62.13%), netilmicin (60.48%), trimethoprim–sulfamethoxazole (58.09%), levofloxacin (57.35%), ciprofloxacin (56.25%), aztreonam (55.70%), and cefpodoxime (50.55%), and confers the least resistance towards cefixime (23.16%) and amikacin (3.86%) (Figure 3). The antibiotic resistance pattern of *E. coli* over time (Appendix A) illustrates its constant elevated resistance towards trimethoprim–sulfamethoxazole, cefuroxime, and ceftazidime, while its overall resistance to cefixime mostly remains below 20%. However, it maintained low resistance levels against amikacin throughout the year.

#### 2.3.2. *Klebsiella* spp.

*Klebsiella* spp. showed low resistance to amikacin (6.99%) (Figure 3), with resistance varying from 0% to 15.79% throughout our study period (Appendix A). In contrast, the isolate exhibited nearly 50% resistance to netilmicin (48.95%), trimethoprim–sulfamethoxazole (48.25%), cefuroxime (46.85%), and ceftazidime (46.85%), along with noticeable resistance towards ampicillin, ciprofloxacin, and levofloxacin (Figure 3). Additionally, the resistance rate of *Klebsiella* spp. (Appendix A) shows that it consistently maintained at least 40% resistance to trimethoprim–sulfamethoxazole, cefuroxime, and ceftazidime throughout most months of the year.

#### 2.3.3. *Enterobacter* spp.

*Enterobacter* spp. projected high resistance against netilmicin (73.53%), and moderately high resistance against ceftazidime (55.88%) and cefuroxime (50.00%). In contrast, it showed comparatively lower resistance to aztreonam (29.41%), cefpodoxime (26.47%), and amikacin (14.71%) (Figure 3). Despite considerable fluctuations in the resistance rate of *Enterobacter* spp. (Appendix A), it exhibited significant resistance to netilmicin, ceftazidime, and cefuroxime, and was generally susceptible to amikacin during most of the months.

#### 2.3.4. *Enterococcus* spp.

*Enterococcus* spp. was the prevalent Gram-positive uropathogen in our study. *Enterococcus* spp. demonstrated more than 70% resistance to trimethoprim–sulfamethoxazole (89.74%), amikacin (75.64%), and ceftazidime (74.36%), while showing the least resistance to cefixime (14.10%) (Figure 3). Throughout most months, *Enterococcus* spp. exhibited high resistance (at least 60%) to trimethoprim–sulfamethoxazole, amoxicillin–clavulanate, ceftazidime, and amikacin (Appendix A). An upwards trend in resistance to ciprofloxacin and levofloxacin was observed, increasing from 33% to 100% over the year, with a slight decrease in October. Except for a few of the months, *Enterococcus* spp. consistently exhibited less than 20% resistance to cefixime.

Overall, our results suggest that most isolates were significantly resistant against ceftazidime, cefuroxime, trimethoprim–sulfamethoxazole, amoxicillin–clavulanate, and netilmicin. Moderate levels of resistance were observed for levofloxacin, ciprofloxacin, aztreonam, and cefpodoxime. While the predominant Gram-negative uropathogens (*E. coli*, *Klebsiella* spp., and *Enterobacter* spp.) exhibited about 85% to 96% sensitivity to amikacin, the major Gram-positive uropathogen, *Enterococcus* spp., showed 85.90% susceptibility to cefixime.

### 2.4. Principal Coordinate Analysis (PCoA) of Antibiotic Resistance and Patient Factors

In order to assess any relationships between the antibiogram and patient clinical data, these antibiotic resistance/susceptibility data were then subjected to a principal coordinate analysis (PCoA) (Figure 4), where the Hamming distances between the points on the plot are close to original dissimilarities between the subjects. The figures here show the same PCoA scores based on the resistance profiles but have been encoded to show the differences in terms of UTI bacteria (Figure 4A), patients’ gender (Figure 4B), and age (Figure 4C), as well as whether they were an inpatient or outpatient (Figure 4D). It can be seen in these plots that the patient’s gender, age, and in- or outpatient status show no correlation with the antibiogram pattern. In addition, Figure 4A shows that there is also a lack of clustering according to antibiotic resistance and susceptibility with the causal UTI bacteria. This is interesting, as this shows that all antibiotics have variable levels of resistance/susceptibility which are not specific to a group of bacteria.

## 3. Discussion

UTIs are extensively encountered in clinical practise and have emerged as a global health problem, occurring as both community-acquired and nosocomial bacterial infections [1,3,7,17].

### 3.1. Higher Prevalence of UTIs among Females

Our study indicated that UTIs were more prevalent among females than males, which is in accordance with the findings of similar reports [6,7,13,17]. The anatomical differences between male and female genitourinary systems, such as the shorter urethra in women, combined with various host factors—including changes during sexual maturation, alterations in normal vaginal flora, pregnancy, and childbirth—likely contribute to the higher incidence of infections among females [17]. Furthermore, a study by Al-Badr and Al-Shaikh [18] states that approximately 50 to 60% of females experience a UTI at least once in their lifetime, especially the elderly.

### 3.2. Age-Related UTI Susceptibility in Men and Women

In our study, the incidence of UTI was highest in women aged 50 to 60 years, reflecting the impact of menopausal alterations on urinary tract health. Post-menopausal women are particularly susceptible to UTIs due to factors like pelvic prolapse, less lactobacilli in the vaginal flora, increased proliferation and colonization of *E. coli* in the periurethral area, decreased estrogen levels, and an elevated prevalence of medical conditions [18]. Men of 60 to 80 years were found to be more prone to UTIs, which may be linked to the elevated incidence of bladder outlet obstructions, like prostatitis and bladder stones, leading to less frequent urination [19]. Additionally, females aged 20 to 50 years exhibited higher infection rates than males, possibly since this age range encompasses the female reproductive years, during which pregnancy is known to increase the risk of UTI [20].

### 3.3. UTI Incidence and Polymicrobial Infection in Pediatric Populations

We also observed that 12.59% of UTI cases occurred in children aged 0–10 years. This finding aligns with other research highlighting significant UTI rates in children, for instance, studies in India [21], Nigeria [22], and Finland [23] reported 10.8%, 9%, and 1 to 5% of UTI cases in children, respectively. However, prolonged hospital stays might contribute to UTIs in children, particularly those caused by extended-spectrum β-lactamase (ESBL)-producing *E. coli* and *Klebsiella* spp. [24]. Furthermore, a significant proportion of patients with polymicrobial UTI were identified to be children compared to the other age groups. Robinson et al. [25] suggests that polymicrobial growth in pediatric UTI cases may result from contamination during urine sample collection. This is often due to the challenges of collecting uncontaminated samples from non-toilet-trained children, who may require methods such as suprapubic aspiration, urethral catheterization, pediatric urine collection bags, or the collection of clean-catch urine while the child is without a diaper. Additionally, contamination frequently occurs in the process of obtaining urine samples from uncircumcised male children due to the challenges associated with retracting the foreskin [25].

### 3.4. Fungal UTIs

In the present study, 5.79% of the total UTI cases were due to fungi, specifically *Candida* spp. Similar surveillance studies conducted in Iran recorded a 5% to 7% prevalence of UTIs associated with *Candida* spp., particularly *Candida albicans* [26,27]. These fungi are prominent opportunistic pathogens responsible for nosocomial UTIs [28]. A total of 51 out of 58 UTI cases caused by *Candida* spp. in our analysis were from hospitalized patients, the majority of whom were female. *Candida albicans* and non-*Candida albicans Candida* spp. are commonly present in the alimentary canal, oral cavity, and vagina of healthy individuals as part of the microflora. In premenopausal and healthy women, these microorganisms inhabit the external urethral opening. Immune deficiencies can disrupt the equilibrium between *Candida albicans*, non-*Candida albicans Candida* yeasts, and the host’s other normal flora. Such conditions trigger the transition of commensal *Candida* yeasts into opportunistic pathogens, leading to UTIs [29].

### 3.5. Gram-Positive Uropathogenic Bacteria

In terms of bacterial uropathogens, 10.32% of the isolates were Gram-positive, predominantly consisting of *Enterococcus* spp. and smaller numbers of *Staphylococcus aureus* and *Lactobacillus* spp. Although lactobacilli are part of the commensal human flora, the prior literature has identified *Lactobacillus delbrueckii* as a causative microorganism in UTIs affecting both genders [30,31]. A cross-sectional study by Malmartel et al. [32] in a French primary care reported *Enterococcus* spp. to be responsible for 7% of 1119 UTI cases, which is consistent with our findings. *Enterococcus* spp., which primarily causes complicated UTIs [1], contributes to over 30% of nosocomial infections due to its ability to survive on various surfaces for extended durations, rendering it the second most common pathogen responsible for catheter-associated UTIs [33]. While *Enterococcus* spp. showed high resistance to trimethoprim–sulfamethoxazole, amikacin, and ceftazidime in our study, cefixime emerged as the most effective antibiotic against it.

### 3.6. Gram-Negative Uropathogenic Bacteria

Of the bacterial isolates identified in our analysis, 89.68% were Gram-negative, consistent with findings from other studies [6,9,17]. The majority of these were *Enterobacterales*, including *E. coli*, *Klebsiella* spp., *Enterobacter* spp., *Proteus* spp., *Citrobacter* spp., and *Morganella morganii*. *Enterobacterales* possess specialized mechanisms for adhering to and colonizing the uroepithelium. These mechanisms include pili, adhesins, fimbriae, and receptors like the P-1 blood group phenotype, which facilitate their attachment to and persistence within the urogenital mucosa. These factors collectively strengthen *Enterobacterales* to cause UTIs, leading to their prominence among bacterial uropathogens [34].

### 3.7. Prevalence of E. coli and Klebsiella *spp.* as Predominant Uropathogens

In our study, *E. coli* was found to be the predominant etiological agent of UTIs, corroborating findings from previous studies that also identified *E. coli* as the principal uropathogen [6,7,9,13,16,17,20,35]. Furthermore, *Klebsiella* spp. was observed to be the second-most common uropathogen in this study, which also aligns with the findings reported by other researchers [6,14,17,36,37]. According to the WHO, *E. coli* and *Klebsiella* spp. are responsible for nearly 80% of UTI cases [38]. In the present analysis, they together accounted for 82.47% of the unimicrobial bacterial uropathogens. *E. coli* and *Klebsiella* spp. can cause both complicated and uncomplicated UTIs. In uncomplicated UTIs, the infection occurs in the lower urinary tract, where bacteria bind directly to the main protein components of the umbrella cell apical membrane in the bladder epithelium (uroepithelium) called uroplakins. These proteins protect mammalian bladder tissue from damaging agents in urine by forming a crystalline array. In contrast, complicated UTIs involve the upper urinary tract and occur when bacteria bind to kidney or bladder stones, urinary catheters, or when they are retained in the urinary tract due to a physical obstruction [1].

### 3.8. E. coli and Klebsiella *spp.*—Resistance towards Cephalosporins and Multidrug Resistance

Among the total bacterial uropathogens, 71.19% were MDR, and 84.27% of the isolates were resistant to at least one of the 11 antibiotics highlighted in this study. This is concerning because a similar study conducted in Bangladesh also reported 78.2% of uropathogens to be MDR [36], while studies in Nepal found multidrug resistance in 54% [35] and 41% [39] of UTI cases. In our study, 73.90% of *E. coli* and 48.95% of *Klebsiella* spp. possessed multidrug resistance. Our finding that *E. coli* has higher rates of MDR than *Klebsiella* spp. is consistent with studies reported in the literature [36,40,41,42,43,44]. Moreover, the antibiotic resistance profile shows 64.71% of *E. coli* and 46.85% of *Klebsiella* spp. confer resistance to cefuroxime and ceftazidime, which are second- and third-generation cephalosporins, respectively. This is particularly concerning because antibiotics of this class are frequently opted for as empiric therapy for patients with severe infections that necessitate hospitalization or intravenous administration [45]. The antibiotic resistance rate of these two uropathogens suggests a reduced efficacy of cefuroxime and ceftazidime in treating UTIs caused by *E. coli* and *Klebsiella* spp., regardless of seasonal or temporal variations. According to two other studies conducted in Bangladesh, *E. coli* and *Klebsiella* spp. exhibited 78.81% and 63.64% resistance to cefuroxime, respectively, in a teaching hospital [7], and 100% resistance to this antibiotic in a tertiary care hospital [14]. Several investigations also indicate *E. coli* and *Klebsiella* spp. show limited sensitivity to ceftazidime. For example, a study conducted in Nepal found *E. coli* and *Klebsiella* spp. to be 82.8% and 96.0% resistant, respectively [35]. Additionally, Sugianli et al. [38] reported in a systematic review (compiling laboratory-based surveillance reports from Australia, Bangladesh, India, and the Republic of Korea) that the prevalence of resistance of *E. coli* and *Klebsiella* spp. to ceftazidime ranges between 2.34% and 68.44%.

However, the elevated resistance against cefuroxime and ceftazidime of *E. coli* and *Klebsiella* spp. is possibly due to their possession of acquired plasmids encoding ESBLs. ESBLs hydrolyse the β-lactam ring inactivating β-lactam antibiotics. Apart from spreading resistance to third-generation cephalosporins, these plasmids rapidly spread resistance against other antibiotics as well, rendering the bacteria to be MDR. This is because the ESBL-encoding plasmids generally also contain genes for resistance against aminoglycosides, sulfonamides, and quinolones [1]. Furthermore, *E. coli* and *Klebsiella* spp. are responsible for the presence of ESBLs in other *Enterobacterales*. This is because incautious administration of cephalosporins in hospital settings and transferrable elements bearing ESBL-encoding genes together create a suitable setting for antibiotic resistance selection in bacteria [1]. Therefore, the careless and excessive misuse of antibiotics in the region is likely to cause this rapid emergence of cephalosporin resistance [14].

In addition to cefuroxime and ceftazidime, we found 44.06% to 62.13% of *E. coli* and *Klebsiella* spp. to exhibit resistance to trimethoprim–sulfamethoxazole, amoxicillin–clavulanate, netilmicin, and ciprofloxacin. A similar finding has been reported by Kothari and Sagar [46] in India, where uropathogens exhibited high levels of resistance to trimethoprim–sulfamethoxazole, amoxicillin–clavulanate, and ciprofloxacin. Additionally, several studies across Asia have consistently documented a substantial resistance of *E. coli* and *Klebsiella* spp. to these antibiotics [47,48,49,50]. The widespread resistance observed may be attributed to the frequent use of these antibiotics as first-line or empirical treatments for uncomplicated UTIs, given their broad spectrum of activity, favourable pharmacokinetics, and general tolerability [38,51].

### 3.9. Susceptibility of E. coli and Klebsiella *spp.* to Amikacin

We found that 96.14% of *E. coli* and 93.01% of *Klebsiella* spp. showed sensitivity to amikacin, and this was the case throughout most months of the year. Our findings are consistent with studies from South Korea, where 0.7% of *E. coli* and 3.4% of *Klebsiella* spp. were amikacin resistant [49]; Iran, where 1–5% *E. coli* and 13–37% *Klebsiella* spp. were amikacin resistant [16]; India, where 0% of both the uropathogens were amikacin resistant [52]; and Bangladesh, where 8.8% of *E. coli* and 13.2% of *Klebsiella* spp. were amikacin resistant [36]. In addition to *E. coli* and *Klebsiella* spp., the majority of uropathogens in this study were observed to be sensitive to this antibiotic, aligning with similar findings by Setu et al. [17] and Mohapatra et al. [37]. The remarkable effectiveness of amikacin against the uropathogens among the tested antimicrobials may be attributed to its uncommon use in empirical UTI treatment, as successful amikacin therapy requires precise patient selection and close monitoring of renal function and potential toxicity [53].

### 3.10. Contributing Factors to the Soaring Antibiotic Resistance

The higher percentages of antibiotic resistance to potent antibiotics observed in our study can be linked to several key risk factors contributing to the rise of antibiotic resistance. A significant factor is the excessive consumption of antibiotics, facilitated by their widespread availability as over-the-counter drugs that can be purchased without a doctor’s prescription. Since Bangladesh has one of the highest numbers of unlicensed stores dispensing antibiotics, even for mild ailments [54], the easy access, combined with their relatively low cost and the convenience of oral administration, has led to inappropriate self-medication and the misuse of antibiotics. Additionally, uropathogens in this study exhibited high resistance to cephalosporins, which are commonly prescribed antibiotics for various infections in Bangladesh [55]. Consequently, factors such as improper dosing, unnecessary prescriptions of effective antibiotics to patients who do not require them, and frequent use of antibiotics for various infections are contributing to the rising antibiotic resistance in local hospitals and the community. Nevertheless, external factors like gender, patient status, or age group did not affect the resistance profile of the bacterial isolates in our study, as deduced from the lack of obvious clustering trends observed in the PCoA plots. Additionally, the PCoA highlighted that the level of resistance or susceptibility is not bacteria-specific and cannot be used to guide general antibiotic stewardship.

### 3.11. Limitations of the Study

Our study has several limitations. Firstly, reliance on retrospective data from a single hospital may limit the generalizability of the findings to other regions. Secondly, detailed information on patient comorbidities or whether the inpatients were catheterized was not documented. This lack of data may have resulted in the inclusion of cases of colonization or asymptomatic bacteriuria. Thirdly, the analysis primarily focused on the predominant bacterial isolates, which may have led to the underestimation of the significance of uropathogens isolated in smaller numbers to ensure reliable data interpretation. Finally, as antimicrobial resistance (AMR) patterns evolve over time, the data from 2018 may not fully reflect current AMR trends and treatment strategies. Additionally, the study’s duration of one year was slightly compromised, as data for August 2018 could not be collected.

## 4. Materials and Methods

### 4.1. Sample Collection

This retrospective study was conducted at Uttara Adhunik Medical College and Hospital (UAMCH) in Dhaka, Bangladesh. Antibiogram data were conveniently collected from hospital records over a one-year period, from January 2018 to December 2018. All patients suspected to have a UTI, regardless of gender (male or female) and age (0 to 100 years), from both outpatient and inpatient departments, were included in this study, except for those already on antibiotic treatment. A total of 6549 samples were collected during this period. UTI was diagnosed depending on microscopic findings of >5 pus cells per high-power field (1000× for high power) and a colony count of ≥10^5^ CFU/mL of one pathogen. In cases where three or more microorganisms were identified at ≥10^5^ CFU/mL and no single species predominated, the infection was classified as a polymicrobial UTI. A sterile, leak-proof, wide-mouthed container was provided to the patients by the laboratory to collect a clean-catch midstream urine sample. Within 1 h of the sample collection, they were sent to the laboratory and processed within 2–4 h at room temperature.

### 4.2. Sample Processing

A standard bacteriological loopful (0.01 mL) of each urine sample was spread over the surface of sterile cystine lactose electrolyte-deficient (CLED) agar plates and left on the bench for some time in order to allow the urine to dry onto the agar medium. The plates were then incubated at 37 °C for 18–24 h in an inverted position. A significant bacterial count was taken as any count equal to or in excess of 10^5^ CFU/mL. Positive samples for both significant bacterial colonies and pus cell tests were cultured aseptically on sterile MacConkey agar, blood agar, and chocolate agar plates, along with other conventional biochemical tests that were performed to identify pure isolates [56,57]. The inoculated plates were incubated at 37 °C for 24 h. Following that, all the identified isolates were subjected to antibiotic sensitivity testing, apart from the fungal isolates.

### 4.3. Antimicrobial Susceptibility Testing

Antimicrobial susceptibility testing of the isolated uropathogens were performed by the Kirby–Bauer disc diffusion technique [56]. The inoculum of the uropathogen was streaked onto sterile Mueller Hinton agar plates aseptically using a sterile inoculating wire loop. Prior to that, the turbidity of the inoculum was adjusted to the 0.5 McFarland standard or 10^8^ cells/mL, according to the recommendations of the Clinical Laboratory Standard Institute (CLSI) [58].

The appropriate antibiotic discs containing amikacin (AK; 30 µg), amoxicillin–clavulanate (AMC; 30 µg), aztreonam (ATM; 30 µg), cefixime (CFM; 5 µg), cefpodoxime (CPD; 10 µg), ceftazidime (CAZ; 30 µg), cefuroxime (CXM; 30 µg), ciprofloxacin (CIP; 5 µg), levofloxacin (LEV; 5 µg), netilmicin (NET; 30 µg), and trimethoprim–sulfamethoxazole (SXT; 25 µg) were then aseptically impregnated onto the surface of the dried plates using sterile forceps. The antibiotic discs used in the study were all obtained from Oxoid Ltd. (Oxoid, Ogdensburg, NY, USA). The plates were left at room temperature for 1 h to allow for the diffusion of the different antibiotics from the disc into the medium. All plates were then incubated at 37 °C for 18–24 h. The results were interpreted depending on the diameter of the inhibition zone according to the CLSI guidelines [58].

### 4.4. Statistical Analysis

These data were collected and analyzed using Microsoft Excel 2007 and MATLAB2019b (MathWorks, Natick, MA, USA). For each sample, a multivariate resistant profile to all the above-mentioned antibiotics was recorded, where ‘1’ indicates resistance to a particular antibiotic and ‘0’ represents susceptibility to this antibiotic. The Hamming distance was used to measure the dissimilarity between each pair of resistant profiles and a principal coordinate analysis (PCoA) was applied to square-rooted Hamming distances to visualize the pattern of these profiles [59].

## 5. Conclusions

In conclusion, 6549 urine samples were analyzed in this study, identifying an alarming prevalence of MDR uropathogens among patients with UTI in a tertiary care hospital in Bangladesh, with *E. coli* and *Klebsiella* spp. being the most prevalent. An elevated resistance of these uropathogens was observed against cephalosporins and other antibiotics commonly used for UTI treatment. However, we found amikacin to be the most effective antibiotic for treating UTIs caused by *E. coli* and *Klebsiella* spp. Although demographic factors such as gender, age, and patient status did not significantly influence AMR patterns of the bacterial uropathogens, our findings suggest continuous surveillance and strategic antibiotic stewardship is inevitable to manage resistance trends effectively and guide empirical treatment strategies, thereby addressing the rise of MDR bacteria and improving treatment outcomes for UTIs.

## Figures and Tables

**Figure 1 antibiotics-13-00925-f001:**
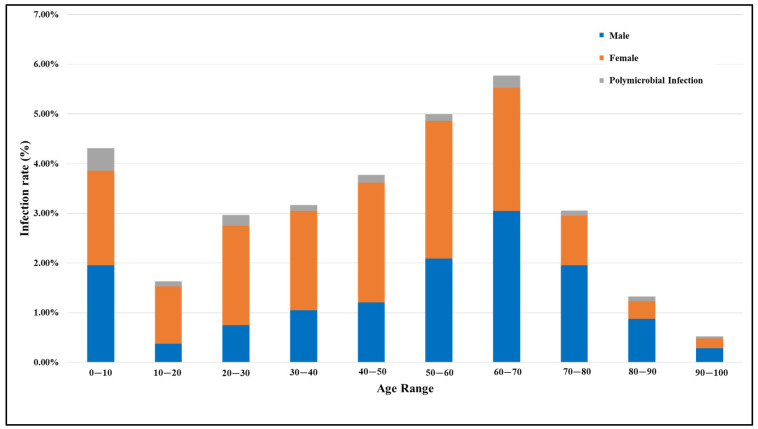
Comparison of UTI rates in males and females across different age groups, including rates of polymicrobial UTIs (infections involving multiple different types of microorganisms).

**Figure 2 antibiotics-13-00925-f002:**
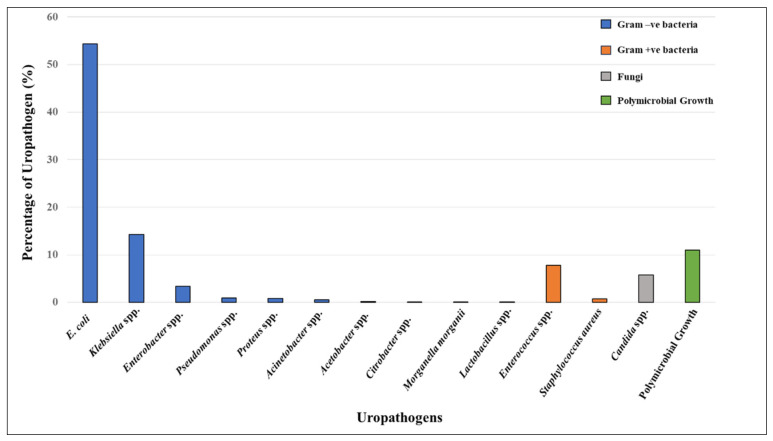
Distribution of uropathogens isolated from patients positive for UTIs (*n* = 1001). The isolates include Gram-negative bacteria (blue bars)—*E. coli*, *Klebsiella* spp., *Enterobacter*, *Pseudomonas, Proteus*, *Acinetobacter*, *Acetobacter*, *Citrobacter*, and *Morganella morganii*; Gram-positive bacteria (orange bars)—*Lactobacillus* spp. *, *Enterococcus*, and *Staphylococcus aureus*; fungi (grey bar)—*Candida* spp.; and polymicrobial growth ** (green bar). * Although lactobacilli are part of the commensal human flora, it can also be a uropathogen in rare instances. ** Polymicrobial growth refers to multiple different types of microorganisms growing together; however, these were not identified in the clinical microbiology lab.

**Figure 3 antibiotics-13-00925-f003:**
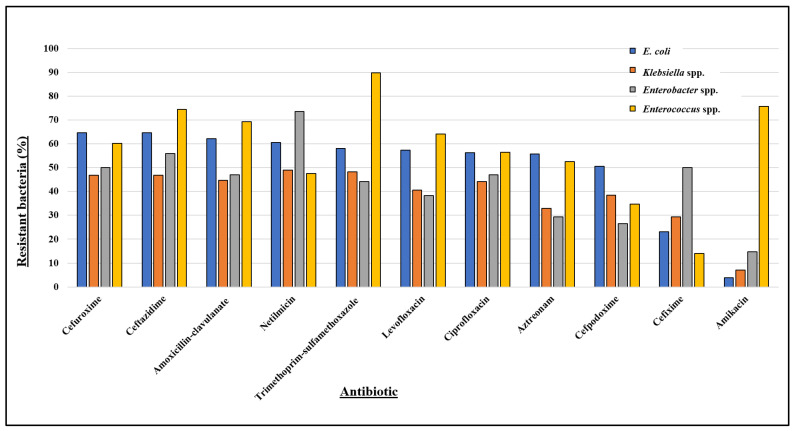
Antibiotic resistance profile of most prevalent bacteria isolated from patients positive for UTIs, where *E. coli* (*n* = 544), *Klebsiella* spp. (*n* = 143), and *Enterobacter* spp. (*n* = 34) are Gram-negative and *Enterococcus* spp. (*n* = 78) is Gram-positive.

**Figure 4 antibiotics-13-00925-f004:**
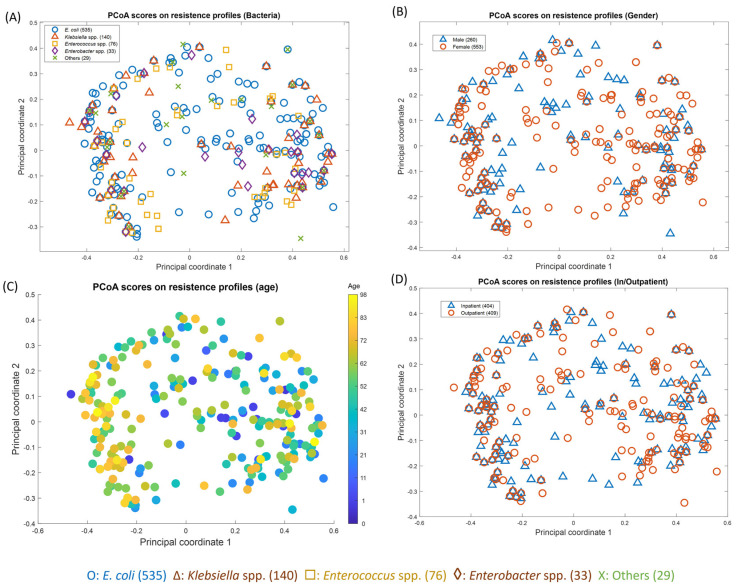
Principal coordinate analysis (PCoA) scores on resistance profiles of *E. coli* (*n* = 544), *Klebsiella* spp. (*n* = 143), *Enterococcus* spp. (*n* = 78), *Enterobacter* spp. (*n* = 34), and others * in terms of (**A**) bacteria, (**B**) gender, (**C**) age, and (**D**) patient status (inpatient/outpatient). * Including: *Pseudomonas* spp. (*n* = 9), *Proteus* spp. (*n* = 8), *Acinetobacter* spp. (*n* = 5), *Acetobacter* spp. (*n* = 2), *Citrobacter* spp. (*n* = 1), *Lactobacillus* spp. (*n* = 1), and *Morganella morganii* (*n* = 1).

**Table 1 antibiotics-13-00925-t001:** Age and gender-specific infection rates of urinary tract infection (UTI).

	Total UTI Suspected Cases (*n* = 6549)		Infection Rate ^1^ (%)	
	Male	Female		
	2392	4257		
	Total UTI Cases (*n* = 1001)			
Age Range	Male	Female	Male	Female
0–10	47	79	1.96%	1.90%
10–20	9	48	0.38%	1.15%
20–30	18	83	0.75%	2.00%
30–40	25	83	1.05%	2.00%
40–50	29	100	1.21%	2.41%
50–60	50	115	2.09%	2.77%
60–70	73	103	3.05%	2.48%
70–80	47	41	1.96%	0.99%
80–90	21	15	0.88%	0.36%
90–100	7	8	0.29%	0.19%
Total	326	675	13.62%	16.25%

^1^ Infection rate (%) = no. of UTI cases in males or females of that age range/total no. of UTI suspected cases in males or females ×100 (*n* = 6549).

**Table 2 antibiotics-13-00925-t002:** Prevalence of multidrug-resistant (MDR) pathogens causing UTIs.

Uropathogens	Frequency of Culture-Positive Isolates	Frequency of ^1^ MDR Isolates	^1^ MDR Prevalence (%) of the Isolates
*E. coli*	544	402	73.90
*Klebsiella* spp.	143	70	48.95
*Enterobacter* spp.	34	24	70.59
*Pseudomonas* spp.	9	9	100.00
*Proteus* spp.	8	5	62.50
*Acinetobacter* spp.	5	3	60.00
*Acetobacter* spp.	2	1	50.00
*Citrobacter* spp.	1	1	100.00
*Morganella morganii*	1	1	100.00
*Enterococcus* spp.	78	74	94.87
*Staphylococcus aureus*	7	2	28.57
*Lactobacillus* spp.	1	1	100.00
Total	833	593	71.19

^1^ MDR = bacterial isolates resistant to more than two antibiotics were identified as multidrug resistant (MDR).

## Data Availability

Data will be made available upon request.

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
