# Peer review of "Trends in Antimicrobial Resistance of Uropathogens Isolated from Urinary Tract Infections in a Tertiary Care Hospital in Dhaka, Bangladesh"

_antibiotics, 2024, doi:10.3390/antibiotics13100925_

Round 1
Reviewer 1 Report
Comments and Suggestions for Authors
Comment
The substantial sample size, a notable strength of this study, significantly enhances the validity of the results. This should instill a sense of confidence in the researchers. However, providing a clear description of the sample collection period is imperative. For a cross-sectional study, specifying the duration of the study period is essential, particularly to ensure that it encompasses at least a year. This would help account for seasonal or temporal variations that might influence the findings.
The microorganism findings include both bacteria and yeast. Lactobacillus, typically a commensal bacterium in the human microbiota, is well-known for its probiotic properties, particularly in maintaining urogenital health by preventing the recurrence of pathogenic bacteria. However, there is documented evidence, including a case report of Lactobacillus delbrueckii causing urinary tract infection (doi: 10.1128/JCM.01630-08), highlighting that certain strains can behave opportunistically under specific conditions. Given this complexity, the authors might want to reconsider revising the term used in Figure 1 to more accurately reflect the dual nature of Lactobacillus as both a commensal and a potential pathogen in rare instances."
Adding data on multidrug resistance patterns or strains would significantly enhance the value of this study. Currently, the manuscript does not present any results on multidrug resistance, which is a crucial aspect in understanding the clinical implications of the findings. Including this information would increase the comprehensiveness of the research and strengthen its relevance and impact within the field, particularly given the growing concern over antibiotic resistance.
It is important to provide the p-values from the statistical analyses to support the finding of no significant differences among the factors in this study. Including these p-values will enhance the transparency and rigor of the statistical evaluation, allowing readers to better assess the results' significance.
In line 96, it is recommended to remove the phrase 'minimum inhibitory' and the abbreviation (MIC) as this study utilized a disk diffusion test. In disk diffusion testing, the concentration of antimicrobial agents on the discs is fixed, and the minimum inhibitory concentration (MIC) is not a relevant measure. Removing these terms will clarify the methodological approach used in the study and help readers better understand the testing procedure.
Describe in brief words Polymicrobial growth in Figure 1.
Please provide a brief description of 'Polymicrobial growth' in Figure 1. This description should clarify that polymicrobial growth refers to multiple different types of microorganisms growing together. Also, if possible, please indicate which microorganisms are involved in the depicted polymicrobial growth.
It would be more beneficial for treatment if the author showed resistance, intermediate, and susceptibility to microorganisms isolated from urine in this study. Moreover, the high percentage of antimicrobial resistance vs. the high strains that were found to be resistant could be added to the abstract.
Author Response
Thank you very much for taking the time to review our manuscript and for your valuable comments, which have greatly contributed to improving the quality of the paper. Please find below our detailed responses to your comments. We have provided two versions of the manuscript: 1) the ‘revised version with track changes’ for review, and 2) the ‘revised version’ of the article with changes made.
Comment 1: The substantial sample size, a notable strength of this study, significantly enhances the validity of the results. This should instill a sense of confidence in the researchers. However, providing a clear description of the sample collection period is imperative. For a cross-sectional study, specifying the duration of the study period is essential, particularly to ensure that it encompasses at least a year. This would help account for seasonal or temporal variations that might influence the findings.
Response: We are pleased that the reviewer likes the scale of the study and its importance. In the previous version of the article, information about the sample collection period was included in Line 64-65 of the Introduction. However, in the resubmitted revised version, this information has been moved to Page 2, Section 2.1 sample collection of the Materials and Methods. It now appears in lines 75-76 as follows: ‘Antibiogram data were conveniently collected from hospital records over a one-year period, from January 2018 to December 2018.’
To account for seasonal and temporal variations, we have included tabular and graphical representations of the resistance rates of the predominant uropathogens throughout the year. These can be found in the supplementary materials file submitted with the revised article. The legends for the table and figure in the supplementary materials are as follows:
- Table S2. Trends in resistance rates of coli, Klebsiella spp., Enterobacter spp. and Enterococcus spp. over time (covering all months of 2018 except August).
- Figure S1. Graphical representation of trends in resistance rates of A) coli ;(B) Klebsiella spp.; (C) Enterobacter spp.; (D) Enterococcus spp. over time (covering all months of 2018 except August).
We have also addressed these points in the following sections: results (lines 179, 192, 200, 204, 235, 245) and discussion (lines 374-376, 408, 440-443).
Comment 2: Lactobacillus, typically a commensal bacterium in the human microbiota, is well-known for its probiotic properties, particularly in maintaining urogenital health by preventing the recurrence of pathogenic bacteria. However, there is documented evidence, including a case report of Lactobacillus delbrueckii causing urinary tract infection (doi: 10.1128/JCM.01630-08), highlighting that certain strains can behave opportunistically under specific conditions. Given this complexity, the authors might want to reconsider revising the term used in Figure 1 to more accurately reflect the dual nature of Lactobacillus as both a commensal and a potential pathogen in rare instances.
Response: The reviewer has made a valid point here. We therefore have modified the figure legend (it is now Figure 2 in the revised article) specifically lines 169-170, to state: ‘Although lactobacilli are part of the commensal human flora, it can also be a uropathogen in rare instances.’
Additionally, we have clarified this point in the results section (lines 170-171), noting that ‘Although lactobacilli are part of the commensal human flora, prior literature has identified Lactobacillus delbrueckii as a causative microorganism in UTIs affecting both genders.’
Comment 3: Adding data on multidrug resistance patterns or strains would significantly enhance the value of this study. Currently, the manuscript does not present any results on multidrug resistance, which is a crucial aspect in understanding the clinical implications of the findings. Including this information would increase the comprehensiveness of the research and strengthen its relevance and impact within the field, particularly given the growing concern over antibiotic resistance.
Response: The reviewer’s point on multidrug resistance is an excellent one. We have now addressed this by adding a new table on page 6 of the revised article, titled "Prevalence of multidrug-resistant (MDR) pathogens causing UTIs," which details the burden of AMR across different bacterial species. Additionally, we have included data on MDR in the abstract (lines 22-25), results (lines 176-178), and discussion (lines 358-359) sections.
Comment 4: It is important to provide the p-values from the statistical analyses to support the finding of no significant differences among the factors in this study. Including these p-values will enhance the transparency and rigor of the statistical evaluation, allowing readers to better assess the results' significance.
Response: Thank you for your suggestion. As this is an observational study, we believe that including p-values may not significantly enhance the analysis. However, we appreciate your comment.
Comment 5: In line 96, it is recommended to remove the phrase 'minimum inhibitory' and the abbreviation (MIC) as this study utilized a disk diffusion test. In disk diffusion testing, the concentration of antimicrobial agents on the discs is fixed, and the minimum inhibitory concentration (MIC) is not a relevant measure. Removing these terms will clarify the methodological approach used in the study and help readers better understand the testing procedure.
Response: Thank you for pointing it out. We have removed it, and the revised article does not contain this phrase anymore.
Comment 6: Describe in brief words Polymicrobial growth in Figure 1. Please provide a brief description of 'Polymicrobial growth' in Figure 1. This description should clarify that polymicrobial growth refers to multiple different types of microorganisms growing together. Also, if possible, please indicate which microorganisms are involved in the depicted polymicrobial growth.
Response: Another good point and we have clarified this, unfortunately no data on the different types of microbes was available to us. We have thus modified the figure legend (it is now Figure 2 in the revised article) specifically lines 172-173, to state: ‘Polymicrobial growth refers to multiple different types of microorganisms growing together; however, these were not identified in the clinical microbiology lab.’
Comment 7: It would be more beneficial for treatment if the author showed resistance, intermediate, and susceptibility to microorganisms isolated from urine in this study. Moreover, the high percentage of antimicrobial resistance vs. the high strains that were found to be resistant could be added to the abstract.
Response: Thank you for your comment. Unfortunately, we do not have specific data on resistance, intermediate, and susceptibility of microorganisms isolated from urine. We have included a summary of the high percentage of antimicrobial resistance and the proportion of resistant strains in the abstract (lines 22-25) of the revised version of the manuscript.
Reviewer 2 Report
Comments and Suggestions for Authors
The manuscript discusses trends in antimicrobial resistance among uropathogens isolated from urinary tract infections (UTIs) in a tertiary care hospital in Dhaka, Bangladesh. The study highlights a significant issue with rising antimicrobial resistance, particularly among common uropathogens such as Escherichia coli and Klebsiella pneumoniae. The results demonstrate an alarming resistance to commonly used antibiotics, including ampicillin, ciprofloxacin, and trimethoprim-sulfamethoxazole.
The comments are given below.
Materials and Methods
Sample Collection
The study was conducted over a one-year period from January to December 2018. Given that antimicrobial resistance patterns are constantly evolving, the data presented may no longer reflect the current situation. This raises concerns about the originality and novelty of the findings. As a result, the study may not effectively evaluate the increase in multidrug-resistant (MDR) bacteria or guide appropriate antibiotic therapy strategies in today's context. I recommend that the authors consider extending the study period to provide more recent and relevant data that could better contribute to understanding MDR trends and informing treatment strategies.
Results
The results are clearly presented, but additional graphs or tables could help visualize some of the findings more effectively. For instance, showing the trends in resistance rates over time in a graph could make the results more accessible.
Discussion
The findings are compared with the existing literature, but expanding this comparison would help to place the study more firmly within the broader research context. For example, brief comparisons with similar studies from South Asia and other regions would further emphasize the study's importance. To improve the quality of the review, the authors may consider expressing this at the discussion.
Line 134: Upon reviewing the data in Table 1, I noticed that the sum of the percentage values for males and females does not match the total percentage given in the rightmost column. I recommend checking and correcting these percentages to ensure accuracy in the data presentation.
Author Response
Thank you very much for taking the time to review our manuscript and for your valuable comments, which have greatly contributed to improving the quality of the paper. Please find below our detailed responses to your comments. We have provided two versions of the manuscript: 1) the ‘revised version with track changes’ for review, and 2) the ‘revised version’ of the article with changes made.
Comment 1: The study was conducted over a one-year period from January to December 2018. Given that antimicrobial resistance patterns are constantly evolving, the data presented may no longer reflect the current situation. This raises concerns about the originality and novelty of the findings. As a result, the study may not effectively evaluate the increase in multidrug-resistant (MDR) bacteria or guide appropriate antibiotic therapy strategies in today's context. I recommend that the authors consider extending the study period to provide more recent and relevant data that could better contribute to understanding MDR trends and informing treatment strategies.
Response: Thank you for your suggestion. While extending the study period is indeed an interesting idea, our study is limited to 2018, and we do not have the data to support an extended period currently. The other two reviewers did not raise this concern. We have acknowledged this as a limitation in the discussion section of the revised manuscript (line 441-443). This could certainly be explored in future research.
Comment 2: The results are clearly presented, but additional graphs or tables could help visualize some of the findings more effectively. For instance, showing the trends in resistance rates over time in a graph could make the results more accessible.
Response: Thank you for your comment. The specific graph you suggested to visualize trends in resistance rates over time is in the supplementary materials titled:
- Figure S1. Graphical representation of trends in resistance rates of A) coli ;(B) Klebsiella spp.; (C) Enterobacter spp.; (D) Enterococcus spp. over time (covering all months of 2018 except August).
We have included the following additional graphs and tables in the revised article titled:
- Figure 1: Comparison of UTI rates in males and females across different age groups, including rates of polymicrobial UTIs (infection involving multiple different types of microorganisms).
- Table 2: Prevalence of multidrug-resistant (MDR) pathogens causing UTIs.
We hope these additions enhance the clarity and accessibility of the findings.
Comment 3: The findings are compared with the existing literature, but expanding this comparison would help to place the study more firmly within the broader research context. For example, brief comparisons with similar studies from South Asia and other regions would further emphasize the study's importance. To improve the quality of the review, the authors may consider expressing this at the discussion.
Response: This is a very useful comment thank you for pointing this out. We have expanded the comparison with existing literature by including studies from South Asia, as well as other regions, in the discussion section. These comparisons can be found in the revised article on lines 282-283, 304-306, 332-334, 350-354, 367-368, 376-379, 380-384, 399-402, 408-412, 423-425. We believe this addition helps to better contextualize our findings within the broader research landscape.
Comment 4: Line 134: Upon reviewing the data in Table 1, I noticed that the sum of the percentage values for males and females does not match the total percentage given in the rightmost column. I recommend checking and correcting these percentages to ensure accuracy in the data presentation.
Response: Thank you for your valuable comment. We realize that the original table could have been clearer in terms of calculations. We have now revised the table to make it more intuitive. You can find the updated version on page 4 of the revised article, titled ‘Table 1. Age and Gender-Specific Infection Rates of UTI’.
Reviewer 3 Report
Comments and Suggestions for Authors
I have read with interest the manuscript submitted by Sara Sadia Chowdhury et al, since AMR represents a global concern.
I have a few comments to be addressed in order to improve the quality of the manuscript:
- each abbreviation should be described in both abstract and main text;
- cefexime - cefixime
Numerous phrases/expressions are incorrect, suggesting that the manuscript needs careful review by an infectionist or clinician. I will give just some examples: "causing 75 to 90% of uncomplicated UTI isolates"; "UTI positive"; "cultures tested positive for UTI"
There is no need to enumerate all antibiotics tested in the aim of the study.
- material and methods - how about the patients with 2 or more microorganisms isolated? were there any urinary-catheterized patients?
Any exclusion criteria?
- Tabe 1 - how about some more advanced statistics to observe significant correlations?
While analyzing the susceptibility profile, Gram-positive bacterium should have been separated from Gram-negative, and include the analysis of specific antibiotics active against each category. Moreover, there is no information about fungal infections (etiology, resistance profile, etc.).
- the discussion section should include more comparisons with other studies;
- replace Enterobacteriaceae with Enterobacterales
The conclusion should emphasize the element of novelty provided by this study. We already know that E. coli is the most common uropathogen and that the AMR pattern varies locally. Delete the general information from the conclusions and focus on the findings of this study
- the reference list is scarce and not edited according to the mdpi pattern
Comments on the Quality of English Language
extensive editing is required.
Author Response
Thank you very much for taking the time to review our manuscript and for your valuable comments, which have greatly contributed to improving the quality of the paper. Please find below our detailed responses to your comments. We have provided two versions of the manuscript: 1) the ‘revised version with track changes’ for review, and 2) the ‘revised version’ of the article with changes made.
Comment 1: Each abbreviation should be described in both abstract and main text.
Response: Thank you for your comment. In the revised article, we have ensured that each abbreviation is described in both the abstract and the main text. Specifically, the full form of urinary tract infection (UTI) is mentioned in the abstract on line 13 and in the introduction on line 40. The full form of antimicrobial resistance (AMR) is mentioned in the abstract on line 16 and in the introduction on line 53. All other abbreviations were already defined in both the abstract and main text.
Comment 2: cefexime – cefixime
Response: Thank you for pointing this out. In the previous version of the article, the misspelling of ‘cefixime’ as ‘cefexime’ occurred in lines 23, 157, and 170. This has now been corrected in the revised article in lines 30, 191, and 248.
Comment 3: Numerous phrases/expressions are incorrect, suggesting that the manuscript needs careful review by an infectionist or clinician. I will give just some examples: "causing 75 to 90% of uncomplicated UTI isolates"; "UTI positive"; "cultures tested positive for UTI"
Response: Thank you for your comment. We have made the following revisions to address this:
- The phrase ‘causing 75 to 90% of uncomplicated UTI isolates’ in line 42 of the previous article has been replaced with ‘accounting for 75 to 90% of bacterial isolates in uncomplicated UTIs’ in line 50 of the revised article.
- The term ‘UTI positive’ in line 151 of the previous article has been revised to “positive for UTI” in line 166 of the revised article.
- The phrase ‘Of 6,549 urine samples, 1,001 (15.29%) cultures tested positive for UTI’ in lines 20-21 of the previous article has been corrected to ‘Out of 6,549 urine samples, 1,001 cultures were positive for UTI’ in lines 19-20 of the revised article.
Comment 4: There is no need to enumerate all antibiotics tested in the aim of the study.
Response: Thank you for your suggestion. We have removed the enumeration as recommended, and the edited lines reflecting this change are 69-71 in the revised article.
Comment 5: How about the patients with 2 or more microorganisms isolated? were there any urinary-catheterized patients?
Response:Thank you for your questions. Patients with 2 or more microorganisms isolated are referred to as having polymicrobial UTIs in our study. The rate of polymicrobial UTIs across different age groups is illustrated in Figure 1 of the revised article: ‘Comparison of UTI Rates in males and females across different age groups, including rates of polymicrobial UTIs (infections involving multiple different types of microorganisms).’
While we were able to extract information about patient status (inpatient/outpatient) from the raw data, we do not have data on whether the inpatients were urinary-catheterized. This limitation has been addressed in the discussion section, lines 436-438 of the revised article.
Comment 6: Any exclusion criteria?
Response: Thank you for your question. Patients who were already undergoing antibiotic treatment were excluded from the study. This is mentioned in the Materials and Methods section, line 79, of the revised article.
Comment 7: Tabe 1 - How about some more advanced statistics to observe significant correlations?
Response: Thank you for your suggestion. Table 1 has been updated to provide a clearer and more intuitive presentation of age and gender-specific infection rates of UTI. We believe that applying additional advanced statistics may not significantly enhance the analysis in Table 1.
On a separate note, we have employed advanced statistics- Principal Coordinate Analysis (PCoA) illustrated in Figure 4 which has not been applied to this type of data before, effectively showing overall correlations.
Comment 8: While analyzing the susceptibility profile, Gram-positive bacterium should have been separated from Gram-negative, and include the analysis of specific antibiotics active against each category. Moreover, there is no information about fungal infections (etiology, resistance profile, etc.).
Response: Thank you for your comments. In the revised article, the susceptibility profiles of Gram-negative and Gram-positive bacteria are analysed separately. Gram-negative isolates are discussed in sections 3.3.1, 3.3.2, and 3.3.3, while Gram-positive isolates are covered in section 3.3.4. Additionally, Table 3 on page 7, titled ‘Antibiotic resistance profile of bacteria isolated from patients positive for UTI,’ has been updated to show the resistance profiles for Gram-negative and Gram-positive bacteria separately.
Fungal infections are addressed in the discussion section, lines 317-328 in the revised article. However, as stated in the materials and methods section, lines 95-96, that all the identified isolates were subjected to antibiotic sensitivity testing, apart from fungal isolates. Therefore, we are unable to provide the resistance profiles of fungal uropathogens.
Comment 9: The discussion section should include more comparisons with other studies.
Response: This is a very useful comment thank you for pointing this out. We have expanded the comparison with existing literature in the discussion section. These comparisons can be found in the revised article on lines 282-283, 304-306, 332-334, 350-354, 367-368, 376-379, 380-384, 399-402, 408-412, 423-425.. We believe this addition helps to better contextualize our findings within the broader research landscape.
Comment 10: Replace Enterobacteriaceae with Enterobacterales.
Response: Thank you for your comment. We have replaced ‘Enterobacteriaceae’ with ‘Enterobacterales’ in lines 342, 344, 347, and 392 of the revised article.
Comment 11: The conclusion should emphasize the element of novelty provided by this study. We already know that E. coli is the most common uropathogen and that the AMR pattern varies locally. Delete the general information from the conclusions and focus on the findings of this study
Response: Thank you for your feedback. The conclusion in the revised article has been revised to emphasize the novel aspects of our study. It now focuses on the key findings and contributions of the study, as detailed in lines 445-456, and avoids general information about commonly known facts.
Comment 12: The reference list is scarce and not edited according to the mdpi pattern
Response: We have revised the reference list to align with the MDPI format as closely as possible. The updated reference list in the revised article now includes 54 references, compared to 28 in the previous version.
Round 2
Reviewer 1 Report
Comments and Suggestions for Authors
Please update the citation in the text. For example, ref. 17 and 19 in the first version of the manuscript, but in the revised version, it might be no 19 and 20. I recommend checking carefully the details of the references, especially CLSI, due to the authors used for the interpretation of antimicrobial susceptibility tests.
For example,
….
…
19. CLSI. Clinical and Laboratory Standards Institute. 2019.
20. Wayne, P. CLSI Performance Standards for Antimicrobial Susceptibility Testing. CLSI Document Clinical Laboratory Standards Institute (CLSI): Wayne, PA, USA 2017.
Moreover, if the laboratory has applied another reference to the zone diameter interpretative criteria, please add it to the method part.
Author Response
Thank you very much for taking the time to review our manuscript and for your valuable comments, which have greatly contributed to improving the quality of the paper. We have taken the opportunity to change the keywords and hope that these changes are acceptable. Please find below our detailed responses to your comments. We have provided two versions of the manuscript: 1) the ‘revised version (with track changes)’, and 2) the ‘final version’ of the article with changes made.
Comment 1: Please update the citation in the text. For example, ref. 17 and 19 in the first version of the manuscript, but in the revised version, it might be no 19 and 20. I recommend checking carefully the details of the references, especially CLSI, due to the authors used for the interpretation of antimicrobial susceptibility tests.
Response: Thank you for pointing this out. We have carefully reviewed and updated the references in the final version of the revised manuscript to ensure accuracy. This includes verifying that reference related to CLSI guidelines is correctly cited in lines 104 and 114. We have ensured that the reference list is consistent with the in-text citations.
Reviewer 2 Report
Comments and Suggestions for Authors
Dear Authors,
Thank you for the revisions made to your manuscript and for addressing the comments provided. Your responses have clarified several aspects of the study, and the added figures and tables indeed enhance the presentation of your results. While you have acknowledged the limitation regarding the study period in the discussion section, extending the study period to include more recent data could offer additional insights into current antimicrobial resistance trends. Future research could explore this to provide updated information.
Best regards
Author Response
Thank you very much for taking the time to review our manuscript and for your valuable comments, which have greatly contributed to improving the quality of the paper.
Reviewer 3 Report
Comments and Suggestions for Authors
I appreciate the author's effort in addressing my comments. The quality of the manuscript has improved.
However, I still have some remarks:
Urinary Tract Infection (UTI) is a prevalent bacterial infection - not only bacterial, although it is the most common etiology;
cultures were positive for UTI - you cannot say that the cultures were positive for uti. Delete "for UTI"
12.59% of total UTI cases occurred - when a sentence starts with numbers, it is advised to write the numbers using letters
Enterococcus species, which is Gram-positive -rephrase, not to state the obvious - active against gram-positive microorganisms, such as Enterococcus spp.
All patients that presented symptoms of UTI - what about urinary catheterized patients?
UTI was diagnosed depending on microscopic findings of > 5 pus cells per high power field (1000x for high power) and a colony count of 105 CFU/mL or greater of one pathogen - how about the polymicrobial UTIs?
Microbiological culture of 6549 urine samples identified 1,001 UTI urine samples identified 1,001 UTI cases (Figure 2). - rephrase
Although lactobacilli are part of the commensal human flora, prior literature has identified Lactobacillus delbrueckii as a causative microorganism in UTIs affecting both genders [22,23] - this phrase does not belong in the results section.
Overall, the majority of the bacterial uropathogens were Gram-negative (89.68%), with E. coli being predominant, followed by Klebsiella spp. and Enterobacter spp. Among Gram-positive bacteria (10.32%), Enterococcus spp. was the most common uropathogen. - avoid duplicate information. this was already mentioned previously.
Figure 2 - Although lactrobacilli are part of the commensal human flora, it can also be a uropathogen in rare instances. **Polymicrobial growth refers to multiple different types of microorganisms growing together; however, these were not identified in the clinical microbiology lab. - again, duplicate information.
Also, why were the microorganisms not identified?
How can the authors explain that E. coli has higher rates of MDR than K. pneumoniae? Usually, K. pneumoniae is associated with higher resistance rates.
On which criteria did you choose only those antibiotics? An individualized study for G+ and G- bacteria would have been more accurate.
table 3 has low resolution. Figure 3 includes the same information from Table 3, it does not bring anything new.
Principal Coordinate Analysis - is not enough as a statistical method. some correlations at least could be calculated.
Any risk factors for AMR?
However, we found amikacin to be the most effective antibiotic for treating UTI caused by E. coli and Klebsiella spp - amikacin alone or in combination with another antimicrobial?
Comments on the Quality of English Languagemoderate editing needed.
Author Response
Thank you very much for taking the time to review our manuscript and for your valuable comments, which have greatly contributed to improving the quality of the paper. Please find below our detailed responses to your comments. We have provided two versions of the manuscript: 1) the ‘revised version (with track changes)’, and 2) the ‘final version’ of the article with changes made.
Comment 1: Urinary Tract Infection (UTI) is a prevalent bacterial infection - not only bacterial, although it is the most common etiology;
Response: Thank you for your comment. We have revised the text in the final version of the revised manuscript to reflect this broader perspective. The term has been updated to ‘Urinary Tract Infection (UTI) is a prevalent microbial infection’ in Line 13 of the abstract.
Comment 2: cultures were positive for UTI - you cannot say that the cultures were positive for UTI. Delete "for UTI"
Response: Thank you for pointing this out. We have removed the phrase as suggested. The text now reads: ‘Out of 6,549 urine samples, 1,001 cultures were positive,’ as updated in lines 19-20 of the abstract.
Comment 3: 12.59% of total UTI cases occurred - when a sentence starts with numbers, it is advised to write the numbers using letters.
Response: We have rephrased the sentence to avoid starting with a number. The updated text now reads: ‘The incidence of UTIs in children aged 0–10 years accounted for 12.59% of the total cases, with this age group also exhibiting the highest rate of polymicrobial infections.’ as revised in lines 20-22 of the abstract.
Comment 4: Enterococcus species, which is Gram-positive -rephrase, not to state the obvious - active against gram-positive microorganisms, such as Enterococcus spp.
Response: Thank you for your feedback. We have rephrased the sentence to avoid stating the obvious. The revised text now reads: ‘Amikacin was observed to be effective against Gram-negative uropathogens, whereas cefixime was more active against Gram-positive microorganisms, such as Enterococcus species.’ as updated in line 31 of the abstract.
Comment 5: All patients that presented symptoms of UTI - what about urinary catheterized patients?
Response: Thank you for your comment. We have revised the information accordingly. The updated text now reads: ‘All patients suspected to have UTI, regardless of gender (male or female) and age (0 to 100 years), from both outpatient and inpatients departments, were included in this study, except for those already on antibiotic treatment. This change is reflected in lines 77-80. Additionally, we have previously acknowledged in our study limitations section that we do not have specific data on catheterized patients.
Comment 6: UTI was diagnosed depending on microscopic findings of > 5 pus cells per high power field (1000x for high power) and a colony count of 105 CFU/mL or greater of one pathogen - how about the polymicrobial UTIs?
Response: Thank you for your question. We have addressed this in lines 82-84. The revised text reads: ‘In cases where three or more microorganisms were identified at ≥105 CFU/mL, and no single species predominated, the infection was classified as a polymicrobial UTI.’.
Comment 7: Microbiological culture of 6549 urine samples identified 1,001 UTI urine samples identified 1,001 UTI cases (Figure 2). - rephrase
Response: Thank you for your suggestion. We have rephrased the sentence to improve clarity in line 154. The revised text now reads: "Microbiological cultures of 6,549 urine samples detected 1,001 cases of UTI (Figure 2)."
Comment 8: Although lactobacilli are part of the commensal human flora, prior literature has identified Lactobacillus delbrueckii as a causative microorganism in UTIs affecting both genders [22,23] - this phrase does not belong in the results section.
Response: Thank you very much for pointing this out. We have moved the sentence to the discussion section, as it is more appropriate there. The revised text can now be found in lines 314-316.
Comment 9: Overall, the majority of the bacterial uropathogens were Gram-negative (89.68%), with E. coli being predominant, followed by Klebsiella spp. and Enterobacter spp. Among Gram-positive bacteria (10.32%), Enterococcus spp. was the most common uropathogen. - avoid duplicate information. this was already mentioned previously.
Response: Thank you for your suggestion. While this information was mentioned in the abstract, it is presented for the first time in the main text in the results section. The total percentages of Gram-negative and Gram-positive bacteria in our study are introduced as new information here.
Comment 10: Figure 2 - Although lactrobacilli are part of the commensal human flora, it can also be a uropathogen in rare instances. **Polymicrobial growth refers to multiple different types of microorganisms growing together; however, these were not identified in the clinical microbiology lab. - again, duplicate information. Also, why were the microorganisms not identified?
Response: Thank you for your suggestion. In the first round of reviews, we were advised by another reviewer to define polymicrobial growth in this figure, which is why it is included here. Additionally, the note that the microorganisms were not identified in the clinical microbiology lab is new information. The microorganisms in the polymicrobial growth were not further identified unless specifically requested by the physician, and in the retrospective raw data they were simply recorded as polymicrobial growth.
Comment 11: How can the authors explain that E. coli has higher rates of MDR than K. pneumoniae? Usually, K. pneumoniae is associated with higher resistance rates.
Response: Thank you for your comment. Our finding that E. coli has higher rates of MDR than Klebsiella spp. aligns with several studies reported in the literature, as referenced in lines 349-351.
Comment 12: On which criteria did you choose only those antibiotics? An individualized study for G+ and G- bacteria would have been more accurate.
Response: Thank you for your question. We selected the antibiotics based on the most frequently used antibiotic disks at the hospital, as reflected in the raw data. The full dataset is available in the supplementary information for further analysis. While we highlighted the antibiotics that were strongly effective against the predominant Gram-negative and Gram-positive bacteria in our study, we believe it is less meaningful to comment on scenarios where only 1% of bacteria were tested with a specific antibiotic, as this would not allow for any sensible conclusions.
Comment 13: table 3 has low resolution. Figure 3 includes the same information from Table 3, it does not bring anything new.
Response: Thank you for your comment. To address this, we have removed Table 3 from the main text and relocated it to the supplementary materials as Table S2.
Comment 14: Principal Coordinate Analysis - is not enough as a statistical method. some correlations at least could be calculated.
Response: Thank you for your suggestion. We introduced Principal Coordinate Analysis to observe associations of the factors in a multivariate space, as it is a novel approach that has not been applied to this type of data before. We think that adding additional methods would not significantly enrich the paper and the other reviewers also did not request for further statistical analysis.
Comment 15: Any risk factors for AMR?
Response: Thank you for your suggestion. We have elaborated on the risk factors for antimicrobial resistance (AMR) in the discussion section, specifically in lines 405-419, and provided relevant references.
Comment 16: However, we found amikacin to be the most effective antibiotic for treating UTI caused by E. coli and Klebsiella spp - amikacin alone or in combination with another antimicrobial?
Response: Thank you for your question. Our study does not evaluate the effectiveness of amikacin in combination with other antimicrobials, as we did not conduct an observational study on combination therapy. However, E. coli and Klebsiella spp. were observed to be most sensitive to amikacin compared to other antibiotics, which is consistent with findings from other studies, as referenced in section 4.9 of the discussion.
Round 3
Reviewer 3 Report
Comments and Suggestions for Authors
I appreciate the author's responses. My last (hopefully) remarks would be:
How were polymicrobial infections differentiated from contamination? This is especially important since this type of infection was more prevalent among children, where the samples are more difficult to collect properly.
I understand that amikacin had the highest sensitivity rates, my remark pointed out that aminoglycosides alone are not indicated in treatment unless no other option is available. Thus, I suggest the authors propose a proper combination of antibiotics, based on the local susceptibility profile.
Regarding the G+/—microorganism tested, I only pointed this out because important antibiotics, such as vancomycin, carbapenems, or ampicillin, are missing. This is important since the AMR levels identified are high.
When asking about risk factors for AMR, I was referring more to the ones identified in your study, not a brief review of some factors identified in other studies.
When reporting antibiotic resistance rates, the EUCAST table with the intrinsic resistance for each microorganism should be consulted.
Moving the table with low resolution in the supplementary data does not solve this issue.
The study's limitation should acknowledge the lack of data on urinary catheterized patients. This means that colonizations/asymptomatic bacteriuria, not only infections, were also included in the analysis.
Best regards,
Comments on the Quality of English Languageminor
Author Response
Many thanks for your careful critique of our manuscript. We have as well as possible answered these final questions. You will see for a couple of these queries we cannot because the information you want is not available. We hope you and the editor will be happy with this and that our manuscript is now acceptable for publication. The other two reviewers are very positive about our work. We have provided two versions of the manuscript: 1) the ‘revised version (with track changes)’, and 2) the ‘final version’ of the article with changes made.
Comment 1: How were polymicrobial infections differentiated from contamination? This is especially important since this type of infection was more prevalent among children, where the samples are more difficult to collect properly.
Response: Thank you for your question. As this study is observational and based on retrospective hospital data, we unfortunately do not have the information on how the hospital differentiated polymicrobial infections from contamination. The hospital documented these samples as having polymicrobial growth. In section 4.3 of the discussion, we indicated that polymicrobial growth was more prevalent among children, which is possibly due to the difficulty in properly collecting urine samples from this population, increasing the risk of contamination.
Comment 2: I understand that amikacin had the highest sensitivity rates, my remark pointed out that aminoglycosides alone are not indicated in treatment unless no other option is available. Thus, I suggest the authors propose a proper combination of antibiotics, based on the local susceptibility profile.
Response: We appreciate your comment. We would like to reiterate that our study is observational, and we do not have the actual data available on the antibiotics that were prescribed. Therefore, we are sorry but not able to provide this information.
Comment 3: Regarding the G+/—microorganism tested, I only pointed this out because important antibiotics, such as vancomycin, carbapenems, or ampicillin, are missing. This is important since the AMR levels identified are high.
Response: Many thanks for the comment. As we have already articulated in the manuscript the details are what we were provided with. This is because not all antibiotics were tested in the assays from all of the cultures. That is to say some initial triaging in the microbiology labs were undertaken.
Comment 4: When asking about risk factors for AMR, I was referring more to the ones identified in your study, not a brief review of some factors identified in other studies.
Response: Many thanks. The risk factors mentioned in the previous version were indeed those identified in our study. The references provided were included to either demonstrate that similar risk factors have been also identified in other studies, or to support key facts about antibiotic use in Bangladesh (the geographical location where our study was conducted). In the revised manuscript, we have specifically highlighted the risk factors identified in our study from lines 405–417. The references in lines 410 and 414, provide supporting information, such as the prevalence of unlicensed stores dispensing antibiotics for mild ailments in Bangladesh and the frequent prescription of cephalosporins in the region.
Comment 5: When reporting antibiotic resistance rates, the EUCAST table with the intrinsic resistance for each microorganism should be consulted.
Response: Thank you for your comment. We can confirm that the data we have used followed EUCAST guidelines.
Comment 6: Moving the table with low resolution in the supplementary data does not solve this issue.
Response: Thank you for your comment. Following an earlier suggestion (that Figure 3 and this table present the same information) we moved the table to the supplementary material. However, we understand that resolution remained to be an issue. We have now ensured that the Table S2. is included in a higher resolution within the supplementary material.
Comment 7: The study's limitation should acknowledge the lack of data on urinary catheterized patients. This means that colonizations /asymptomatic bacteriuria, not only infections, were also included in the analysis.
Response: Thank you for your comment. We agree that the lack of detailed data on catheterization is an important limitation of the study, and we have previously acknowledged the absence of data on catheterization status of the patients in our limitation section. However, we have now ensured that this point is more explicitly stated to clarify the potential influence of this limitation in line 428-429.